# Learning Performance of International Students and Students with Disabilities: Early Prediction and Feature Selection through Educational Data Mining

**Thao-Trang Huynh-Cam** [1,2], **Long-Sheng Chen** [1,*] and **Khai-Vinh Huynh** [3]

1    Department of Information Management, Chaoyang University of Technology, Taichung 413310, Taiwan
2    Foreign Languages and Informatics Center, Dong Thap University, Cao Lanh City 81118, Vietnam
3    Quality Assurance Office, Dong Thap University, Cao Lanh City 81118, Vietnam
*    Correspondence: lschen@cyut.edu.tw

**Abstract:** The learning performance of international students and students with disabilities has increasingly attracted many theoretical and practical researchers. However, previous studies used questionnaires, surveys, and/or interviews to investigate factors affecting students' learning performance. These methods cannot help universities to provide on-time support to excellent and poor students. Thus, this study utilized Multilayer Perceptron (MLP), Support Vector Machine (SVM), Random Forest (RF), and Decision Tree (DT) algorithms to build prediction models for the academic performance of international students, students with disabilities, and local students based on students' admission profiles and their first-semester Grade Point Average results. The real samples included 4036 freshmen of a Taiwanese technical and vocational university. The experimental results showed that for international students, three models: SVM (100%), MLP (100%), and DT (100%) were significantly superior to RF (96.6%); for students with disabilities, SVM (100%) outperformed RF (98.0%), MLP (96.0%), and DT (94.0%); for local students, RF (98.6%) outperformed DT (95.2%) MLP (94.9%), and SVM (91.9%). The most important features were [numbers of required credits], [main source of living expenses], [department], [father occupations], [mother occupations], [numbers of elective credits], [parent average income per month], and [father education]. The outcomes of this study may assist academic communities in proposing preventive measures at the early stages to attract more international students and enhance school competitive advantages.

**Keywords:** international students; students with disabilities; learning performance prediction; educational data mining; technological and vocational education

## 1. Introduction

The learning performance of minority students, such as international students and students with disabilities, has increasingly attracted considerable theoretical and practical attention from researchers and educational teams since it is currently considered to be one of the crucial criteria for assessing campuses' quality [1]. Early prediction of minority students' performance and the determining of important features affecting their learning performance are indispensable to higher education institutions (HEIs). Students' big data analysis can assist academic communities to foresee students' learning conditions, provide on-time support, and propose preventive measures at the early stages before students start the first semester. Thus, HEIs will attract more international students and enhance their competitive advantages in global educational environments.

Studying abroad benefits both international students and host campuses. For students, the extended stays abroad are considered valuable opportunities for acquiring and improving various skills, such as foreign language competence and intercultural skills [2,3]. Vice versa, international students contribute significantly to the financial resources, cross-culture environments, and high-quality human resources (i.e., research and teaching assistants) of

host universities. HEIs' staff can learn various cultures from different countries; schools can have more opportunities for further cooperation with global campuses from alumni recommendations [2,4].

In recent years, compared to Asian countries, Taiwan has offered many advantageous supports in high-quality international academic environments. Thus, thousands of international students have studied in Taiwan during the past decade [2,5,6]. Annually, the numbers of international students in Taiwan, including degree and non-degree level (i.e., exchange and language-study), have increased considerably [2,4,5,7]. Particularly, the populations of international degree-level students increased from 6380 in 2001 to 21,005 in 2007 [5,8]. The numbers of overseas students in Taiwan rose considerably, from approximately 63,000 in 2012 to 130,000 in the 2019 academic year [6]. The incoming international students remain unchanged owing to the COVID-19 pandemic.

Nevertheless, according to Education in Taiwan 2016–2017 [9], there is a huge increasing number of students with special needs in various levels between 2001–2015 in Taiwan. In particular, the rising number of students with disabilities is from 3689 to 15,559 in preschool, from 35,721 to 42,022 in primary, from 20,993 to 28,228 in junior high school, from 6952 to 23,577 in senior and vocational high school, and from 2961 to 12,376 in higher education. The government budget for supporting students with disabilities has increased from NTD$5.579 billion in 2001 to NTD$9.903 billion in 2015.

Owing to the digitalization of academic processes, large data repositories provide a vast educational data (ED) for learning and analyzing how students learn [10]. Data mining (DM) is also called knowledge discovery in databases (KDD). DM is concerned with data analysis by using different techniques/software to extract meaningful information and knowledge from the raw data sources and to identify the relationship among the dataset [11–14]. The DM process comprises data cleaning, data integration, data selection, data transformation, data mining, and knowledge representation [15]. DM techniques including machine learning algorithms and artificial intelligence (AI) have been widely and successfully applied to predict and to identify the important features associated with student academic performance [16].

Educational data mining (EDM), an application of DM in education sectors, has recently become popular and inspired many researchers. EDM can be used to understand students' learning conditions, students' behaviors, and their subject interests in order to improve teaching supports, and to make decisions in educational systems [17,18], since most ED are collected from students' interactions and behaviors [12,19]. EDM's models are capable of forecasting student knowledge and future performance [11]. However, from available research, no published papers focused on predicting and determining factors which impact greatly on the learning performance of international and special-need freshmen in Taiwanese vocational and technical universities.

Therefore, this study utilized EDM techniques to build prediction models for the academic performance of international students, students with disabilities, and local students based on students' admission profiles and their first-semester grade point average results collected from the school database system of a Taiwanese vocational and technical university. The study may benefit stakeholders in the educational sectors. Teachers can adjust their pedagogical strategies and teaching methods when teaching international and/or special-needs students. Policymakers can plan, design, and implement institutional policies to improve the international and special-needs students' performance, propose preventive measures at the early stages before students start the first semester in order to attract more international students, and enhance campuses' competitive advantages in the global education markets. The government can establish educational policies from the analysis of cross-institutional data [12,19]. Parents can solve the students' problems, especially financial problems and/or guide them properly.

## 2. Background and Related Works

### 2.1. Educational Data Mining (EDM)

EDM is data mining in the educational sector. EDM is the process by which the collected raw data from the educational environment based on hypothesis formation and/or educational questions are pre-processed and imported into DM instruments/software to produce models/patterns for interpretation and evaluation [10,12]. The results of interpretation and evaluation are then used to refine the educational environment and hypothesis formation. Correspondingly, Siemens and Baker [20] suggest that EDM is a practice of DM methods for studying big datasets in order to get insights from students and educational systems. EDM is "not only to turn data into knowledge, but also to filter mined knowledge for decision-making about how to modify the educational environment to improve student's learning" [21]. It has been applied to evaluate the pedagogical support of a specific learning tool and recommend potential improvements [21]. The goals of EDM include (1) predicting student's learning behavior; (2) exploring or upgrading domain models; (3) analyzing the effects of various instructional support types; and (4) advancing scientific knowledge [22,23]. To add value, EDM must assist in exploiting the multiple levels of meaningful hierarchy in ED, especially predicting student learning performance and their future achievement for long-term development.

### 2.2. Student Learning Performance Prediction

Prediction is to infer a target attribute (predicted variable) from a combination of other aspects of data (predictor variable) [10,11]. Prediction requires labels for output variables. In EDM, prediction has been popularly used for forecasting student performance and for detecting students' behaviors [10,11]. The category of output variable for prediction can be either categorical or continuous.

Student learning performance prediction (SLPP) has been deeply concerned by HEIs, since early SLPP can lead to careful strategic intervention plans before students reach the final semester [24] and even prevent students at-risk/dropouts by providing them with additional assistance or tutoring on time [25]. Therefore, SLPP has increasingly attracted a diversity of researchers and educational teams in educational sectors. In order to gain a high-value contribution, various prediction techniques and feature selection methods are also utilized.

The main stream of SLPP methods are supervised learning, since "students' academic learning performance" is the class label. Prediction models are implemented by algorithms on specific computational software/tools. Popular algorithms employed for SLPP include artificial neural network (ANN), decision tree (DT), support vector machine (SVM), random forest (RF), k-nearest neighbor (KNN), and Naïve Bayes [26,27]. DT is popularly and successfully applied for prediction and classification in the domain of machine learning due to its simple use, easy understanding, and high prediction accuracy [26–29] through a flowchart-like tree structure and IF-THEN rules [30–32]. In other words, DT is a model-based approach using a tree-shaped graph. It is a root-to-leaf route which represents classification rules [25,33]. In its flowchart-like tree structure, each inner node represents a test on the feature/variable, each branch represents the test result, and each leaf represents a class label [25]. Compared to other methods, DT can extract readable knowledge rules, which is helpful for university-side decision-making references [34]. However, one of the biggest drawbacks of the DT model is overfitting, which can be solved by the presence of RF [25,35,36]. SVM is good for handling small datasets, while ANN can solve the nonlinear and complex relationship between various input and output variables [37,38].

The SLPP features are mainly grouped into popular categories: demographic, academic performance, internal assessment, communication, behavioral, psychological, and family/personal background [38]. For instance, Huynh-Cam et al. [24] measured the impact of family background features on first-year undergraduates using DT and RF. Matzavela and Alepis [31] assessed the effects of gender, grade, parent education, parent income, whether one is the first child or not, and whether a student is working or not on university

student performance using DT; whereas authors in [30] evaluated university students final GPA using J48 DT. Niyogisubizo et al. [39] examined the impact of the features: access, tests, tests grade, exam, project, project grade, assignments, result points, result grade, graduate, year, and academic year on student dropouts in higher education using ANN and RF.

*2.3. Learning Performance Prediction of Minority Students*

International students contribute significantly to host HEIs' diversity, revenue, investment, research, and teaching [2,4,40]. Thus, it is important for the host HEIs' heads to be aware of factors associated with international students' learning performance in order to offer in-time support, prevent at-risk dropout, maintain international students' retention, and attract a larger number of international students. Therefore, this issue has been one of significant concern of policy makers, practitioners in HEIs and researchers in the past decades. In Taiwanese universities, there were three main groups of studies focusing on top-ranking factors associated with international student performance. Group one included simplified Chinese characters, Taiwanese government scholarships, and high-quality Mandarin Studies programs [5]; group two consisted of stressors, living support, and adjustment [2]; and group three comprised social adjustment experiences to campus life [7]. In contrast, for graduate students in the United States, the important factors included gender, age, native region, native language, undergraduate GPA, proportion of time studying alone, teaching and learning methods, and length of study time [40]. In German high schools, socioeconomic background, grades, and course choice in English positively affected studying abroad during high school [3].

Inclusive education for students with special educational needs in regular classrooms has currently become an increasing trend [41]. It is also one of the required criteria for assessing campuses' quality in many countries since it is claimed that students with disabilities should have equal rights to receive education in mainstream schools [41,42]. Therefore, the academic performance of students with special educational needs has attracted many researchers. For instance, Griffiths et al. [43] described the co-development of a six-phase tripartite model for a supportive framework for nursing students with special educational needs through an individual student pathway. Huang et al. [41] analyzed the academic peer influence of students with disabilities in the classroom of Chinese middle schools using students' midterm scores in three compulsory subjects: Chinese, math, and English. Hersh [44] introduced an evaluation framework for ICT-based learning technologies for students with special needs by a systematic approach. Zainudin et al. [45] used a questionnaire to investigate the effects of family support, facility support, and lecture support on special-needs students' academic performance in e-learning during the COVID-19 pandemic.

*2.4. Research Gaps*

From the available published studies, none of them employed EDM approaches/techniques to predict the academic performance of international students and students with disabilities. In addition, among these limited studies, many researchers used questionnaires and/or interviews to investigate factors associated with international and special-needs students' learning performance. These methods cannot help universities provide on-time support to excellent and poor students.

There are a wide variety of SLPP techniques (Section 2.2). However, this study applied four well-known classification techniques: multilayer perceptron (MLP), decision tree (DT), support vector machine (SVM), and random forest (RF) algorithms to build prediction models owing to their critical features. We employed DT owing to its wide application, ease of use, and simplicity. We used MLP and SVM due to the nature of our small datasets, which did not require more complex algorithms. RF can help solve the overfitting/overtraining problems.

## 3. Experimental Methodology

Figure 1 summarizes the five-step experimental process in this study. Step 1 reports data collection methods and describes the research datasets. Step 2 describes a two-step data pre-processing process: data cleaning and data normalization. Step 3 explains a three-step model implementation: data split, feature selection, and model prediction building. Step 4 evaluates model performance and Step 5 concludes extracted knowledge from EDM and the important features affecting minority students' academic performance. The following subsections will explain these five steps in detail.

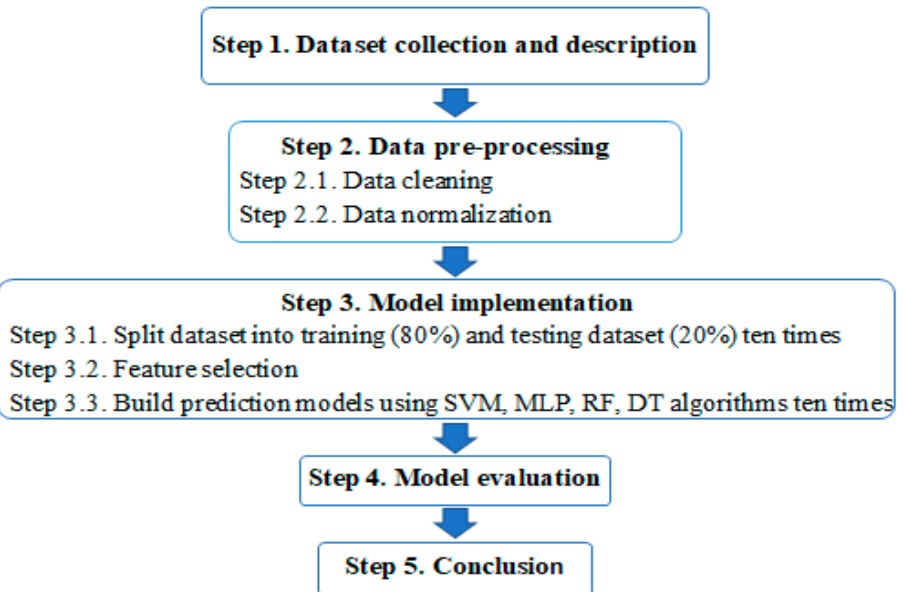

**Figure 1.** Experimental process.

### 3.1. Step 1. Dataset Collection and Description

The raw dataset used to conduct the experimental study was collected directly from the school database system of a Taiwanese technical and vocational university during the first semester of the 2020–2021 academic year. The original dataset contained 7736 en-rolled records of first-year students after official admission. For each student, there were 22 categories of profile information: [department], [gender], [address], [admission status], [aboriginal], [child of new residents], [family children ranking], [parent average income per month], [on-campus accommodation], [main source of living expenses], [student loan], [tuition waiver], [father live or not], [father's occupations], [father's education], [mother live or not], [mother's occupations], [mother's education], [numbers of required credits], [numbers of elective credits], [sick leave], and [personal leave] were used as input variables. The first-semester grade point average (GPA) results in the students' bachelor program was used as an output variable. These students' identity remained anonymous for ethical reasons.

### 3.2. Step 2. Data Pre-Processing

Data pre-processing is a DM technique which is used to convert the original data, raw or primary data into a useful and efficient dataset by addressing DM algorithms [21,46,47]. This phase consists of two steps: data cleaning and data normalization.

Step 2.1: Data cleaning: In this step, irrelevant attributes and missing-value samples were removed. In addition, all category features were encoded and transferred to binary and/or numeric features.

At first, we eliminated some irrelevant attributes, e.g., student name, student ID, birth date, and birthplace. Then from 22 initial features (Section 3.1), we removed four features: [aboriginal], [child of new residents] [mother alive or not], and [father live or not] since our

previous study [24] illustrated that they hardly affected students' learning performance. The feature [address] is inconsistent among the three target groups of students because its corresponding geographic regions are only used for Taiwanese students, e.g., North, South, East, West, and Island. In addition, we deleted the feature [admission status] because we only focused on degree students, not exchange students, transfer students and non-degree students. After deleting irrelevant attributes, out of 22 initial features, a total number of 16 features were finally selected for model building, as listed in Table 1.

**Table 1.** The employed input and output data description and transformation.

| No. | Feature Name | Feature Description and Transferred Values | No. | Feature Name | Feature Description and Transferred Values |
|---|---|---|---|---|---|
| 1 | Department | 1 = TCA, 2 = TCJ, 3 = CK, 4 = TCL, 5 = TDJ, 6 = TDN, 7 = TD4, 8 = TD5, 9 = TD6, 10 = TD7, 11 = TC6, 12 = TC7, 13 = TC8, 14 = TC9, 15 = TE1, 16 = TE2, 17 = TE3, 18 = TE4, 19 = TE5, 20 = TQ1, 21 = TF1, 22 = TJ2, 23 = TJ4, 24 = TF2, 25 = TF3, 26 = TF4, 27 = TJ9 | 9 | Main source of living expenses | 1 = Parents 2 = Family and friends support 3 = Self-earning 4 = Grants in- or outside the school 5 = Income from full-time job 6 = Family provided 7 = Income from part-time job 8 = Scholarships 9 = Student loans |
| 2 | Gender | 1 = Male, 2 = Female | 10 | Student loan | 1 = Yes, 0 = No |
| 3 | Numbers of required credits | 0–23 | 11 | Tuition waiver | 1 = Yes, 0 = No |
| 4 | Numbers of elective credits | 1–14 | 12 | Father's occupations | 1 = Military 2 = Education 3 = Public 4 = Service 5 = Industry 6 = Business 7 = Agriculture 8 = Others |
| 5 | Sick leave | 0–36 | 13 | Father's education | 1 = Junior high school and below 2 = High school 3 = Bachelor 4 = Master 5 = Specialist 6 = PhD |
| 6 | Personal leave | 0–33 | 14 | Mother's occupations | 1 = Military 2 = Education 3 = Public 4 = Service 5 = Industry 6 = Business 7 = Agriculture 8 = Others |
| 7 | Parent Average income per month | 1 = 25,000 NTD, 2 = 40,000 NTD, 3 = 60,000 NTD, 4 = 80,000~100,000 NTD, 5 = Above 100,000 NTD | 15 | Mother's education | 1 = Junior high school and below 2 = High school 3 = Bachelor 4 = Master 5 = Specialist 6 = PhD |
| 8 | On-campus accommodation | 1 = Yes, 0 = No | 16 | Grade Point Average (GPA) | 1 = Excellent (90–100 points), 2 = Very Good (80–89 points), 3 = Good (70–79 points), 4 = Average (60–69 points), 5 = Poor (0–59 points) |

Note: Feature 10 and 11 were not applied for Group 1 (international students).

Features 1–15 were input (independent) variables and Feature 16 was the output (dependent) variable for prediction models. Nevertheless, Feature 10 [student loan] and Feature 11 [tuition waiver] were not applied for Group 1 (international students), since the school did not offer loan and tuition waiver to these students at the research time.

After data cleaning, out of 7736 original examples, a total of 4036 (52.2%) students were finally selected for experimental study. These students were divided into three target groups: international students (Group 1), students with disabilities (Group 2), and local students (Group 3). Figure 2 displays the numbers of employed students in each group. The remaining 3700 (47.8%) students, who had missing values, dropped out, and/or were suspended before implementing the experiment, were excluded from this study.

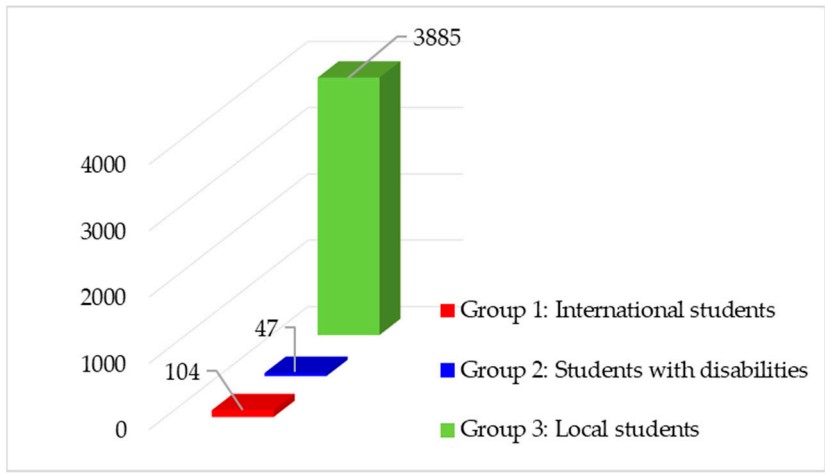

**Figure 2.** Numbers of employed students in each group.

Finally, we encoded and transferred all category features to binary and/or numeric features as described in Table 1. We also constructed categorical target variables based on the original numeric parameter university GPA scores which are in compliance with the Taiwanese grading system (100-points scale). The target variable (Feature 16) has five categories: excellent (90–100 points), very good (80–89 points), good (70–79 points), average (60–69 points), and poor (0–59 points).

Step 2.2: Data normalization. In this step, the data was normalized in accordance with Equation (1), which was applied in our previous study [24].

$$X_{mon} = \frac{X - X_{min}}{X_{max} - X_{min}} \tag{1}$$

where $X_{max}$ is the maximum value, $X_{min}$ is the minimum value, and $X_{mon}$ is the normalized value.

*3.3. Model Implementation*

There are three steps in this phase. In Step 3.1, the input data were randomly divided into training and testing datasets ten times with percentages of 80% and 20%, respectively. Step 3.2 was for feature selection in order to overcome overfitting problems and better prediction performance [37,39,48].

Step 3.3 was to build prediction models. In this step, we employed four supervised classification algorithms: SVM, MLP, RF, and DT in the Python language which were widely used in machine learning. The software tool chosen for building prediction models in this study was Jupyter, an open-source software project affiliated with the 501c3 Num-FOCUS Foundation. The software was developed openly by the Jupyter/IPython Project and was hosted in public GitHub repositories under the IPython GitHub organization (https://github.com/ipython, accessed on 1 February 2022) and the Jupyter GitHub organization (https://github.com/jupyter, accessed on 1 February 2022). Jupyter Project is developed by a team of contributors. Contributors are individuals who have contributed code, documentation, designs or other work to one or more the Project repositories. The Python and Jupyter packages are available at scikit-learn: machine learning in Python—scikit-learn 1.0.2 documentation and https://www.dataquest.io/blog/jupyter-notebook-tutorial

(accessed on 1 February 2022). Each model was experimented on ten times with ten different training-testing datasets. The mean value and standard deviation of the 10-time experiment for each model were taken and used for benchmarking prediction performance among SVM, MLP, RF, and DT models. After selecting the prediction model with the best performance, the Gini index was applied to select important features associated with students' learning performance, especially the minority students: international students and students with disabilities. Furthermore, in order to achieve the highest performance accuracy, we classified the output variable into three cases (Figure 3).

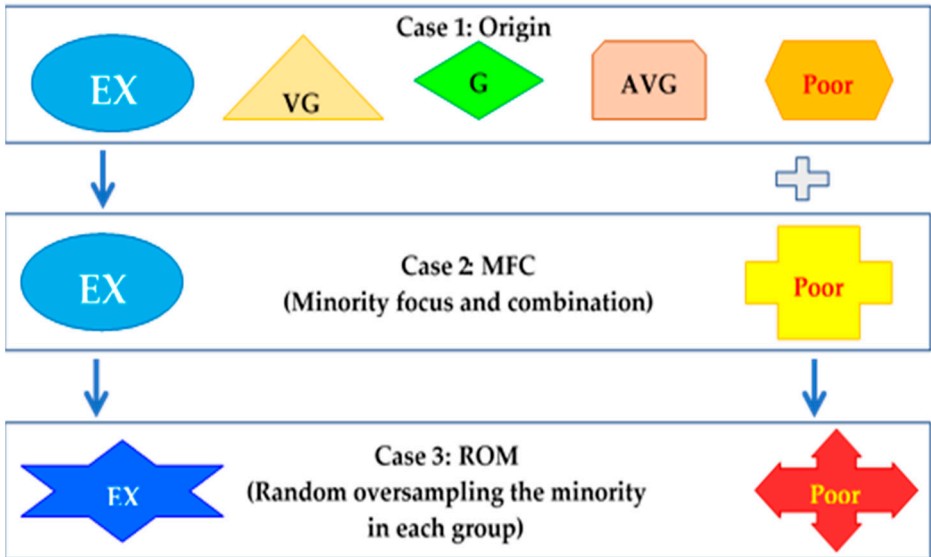

**Figure 3.** Class distribution in three cases. Note: EX = class "Excellent", VG = class "Very good", G = class "Good", AVG = class "Average".

- **Case 1: Origin** included 5 origin classes: Excellent (EX), Very Good (VG), Good (G), Average (AVG), and Poor. This case was used for investigating if the models predict the minority or not. Figure 4 graphically displays the numbers of employed samples in each group.

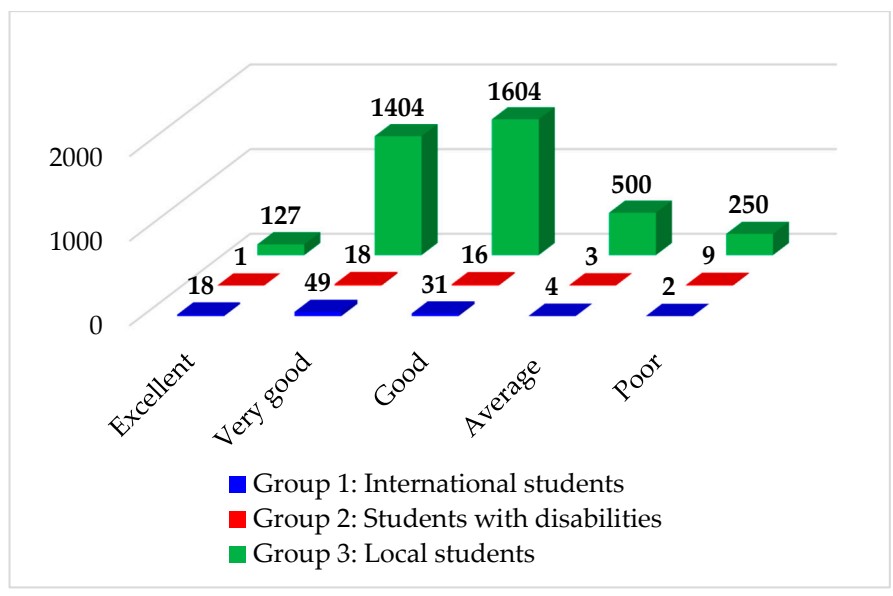

**Figure 4.** Numbers of samples in each group in Case 1: Origin.

- **Case 2: Minority focus and combination (MFC)** focused on two minority classes: EX and Poor. The majority classes VG and G were removed. Since the classes "AVG" and "Poor" were very few, we combined the two minority AVG and Poor classes into the Poor class. Therefore, the new combined Poor class (AVG+Poor) included six samples in Group 1, 12 samples in Group 2, and 750 samples in Group 3 (Figure 5a). However, after combining, the imbalanced data problem was present in each group. As shown in Figure 5a, the Poor (AVG+Poor) class remained the minority class in Group 1 (nPoor = 6; nEX = 18); whereas it became the majority class in Group 2 (nPoor = 12; EX = 1) and Group 3 (nPoor = 750; nEX = 127). Therefore, we proposed a resampling method: random oversampling the minority (Case 3) to solve the imbalanced data problem in each group (Figure 5b).

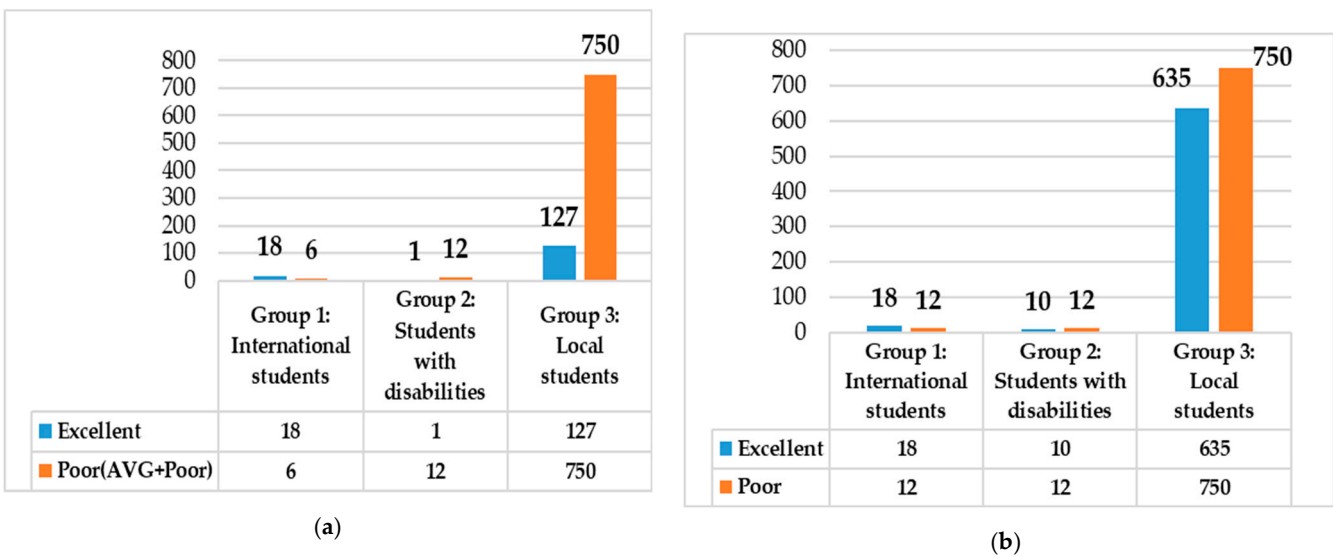

(**a**)    (**b**)

**Figure 5.** Numbers of samples in each group: (**a**) Case 2: MFC; (**b**) Case 3: ROM.

- **Case 3: Random oversampling (ROM)** was to randomly oversample the minority class in each group by duplicating or generating new minority class instances [49,50]: "EX" and "Poor" classes. As shown in Figure 5b, the numbers of samples in each group are approximately balanced. In the works of Chen et al. [49] and Chang et al. [50], they indicated that oversampling is one of effective solutions for tackling class imbalance problems. Therefore, we employed ROM to deal with class imbalance problems in this study.

*3.4. Model Evaluation*

In order to evaluate the effectiveness of our proposed method (ROM), we employed three methods.

Method 1: we benchmarked the prediction accuracy of Case 3 (ROM), Case 2 (MFC), and Case 1 (Origin). However, relying only on accuracy could lead to misinterpretation when the classification involves imbalanced data [49,51], since the models can predict the majority class and ignore the minority class [39]. Therefore, we employed various evaluation metrics.

Method 2: we compared the accuracy (ACC), precision (PR), recall (Rec), the area under the curve (AUC), and F1-score (F1), which were employed by the authors in [39,52–54]. We also used confusion matrix results to explain the classification models. The parameters of the confusion matrix were:

- True Positive (TP): instances, which are actually positive, are classified as positive.
- False Positive (FP): instances, which are actually negative, are classified as positive.
- False Negative (FN): instances, which are actually positive, are classified as negative.

- True Negative (TN): instances, which are actually negative, are classified as negative.

In this study, Positive instances were EX class and Negative instances were Poor class. The formula of all metrics used in the evaluation stage are described below:

$$\text{ACC} = \frac{\text{True Positive (TP)} + \text{True negative (TN)}}{\text{True Positive (TP)} + \text{True negative (TN)} + \text{False Negative (FN)} + \text{False Positive (FP)}} \tag{2}$$

ACC is used to find the portion of correctly classified values.

$$\text{PR} = \frac{\text{TP}}{\text{TP} + \text{FP}} \tag{3}$$

PR is used to calculate the model's ability to classify positive values correctly.

$$\text{Rec} = \frac{\text{TP}}{\text{TP} + \text{FN}} \tag{4}$$

Rec is used to calculate the models' ability to predict positive values.

$$\text{F1} = \frac{2 * \text{Rec} * \text{PR}}{\text{Rec} + \text{PR}} \tag{5}$$

F1-score is the harmonic mean of Recall and Precision, which is used when we need to take both PR and Rec into account.

Method 3: we utilized the area under the curve (AUC), which is frequently used to measure the prediction performance of a classification method for all classification thresholds [39]. The AUC values range between 0.5 and 1.0, where the 1.0 value indicated the excellent performance and the 0.5 value was for the poor performance of a specific model.

## 4. Experimental Results

After pre-processing, the dataset was imported to Jupyter software to implement MLP, RF and DT, and SVM models. About parameter settings in RF, the number of trees in the forest is set to 100. With regard to decision trees, pruning confidence value (CF) affects the way of estimating the error rate, thereby affecting the severity of pruning in order to avoid overfitting of the model. In this study, pruning CF was set to 25%. In MLP, the learning rate is set to 0.3, and the training stop condition is set to the number of learning iterations to 1000. At this time, the RMSE (Root-Mean-Square Error) has been flattened, meaning that the network has converged. All optimal parameter settings of SVM could be obtained automatically by using the grid search technique.

### 4.1. Results of Case 1: Origin

In this case, we used accuracy, precision, recall, and F1-score values to evaluate the prediction performance. Table 2 summarizes all results for three cases. From this table, we can find that all accuracies of the four classifiers, SVM, MLP, RF, and DT are very low. The same situation could also be found in PR, Rec, and F1. In other words, we cannot identify the minority classes "Excellent" and "Poor". If we cannot identify these minority classes, the classification results will be meaningless. Therefore, we implement Case 2, which is minority focus and combination (MFC). In MFC, we only focused on two minority classes, EX and Poor.

**Table 2.** Classification results of Case 1 (Origin).

| Performance | Accuracy (%) | | Precision | | Recall | | F1-Score | |
|---|---|---|---|---|---|---|---|---|
| | Mean | SD | Mean | SD | Mean | SD | Mean | SD |
| Methods | | | Group 1: International students | | | | | |
| SVM | 43.00 | 0.00 | 0.23 | 0.00 | 0.26 | 0.00 | 0.24 | 0.00 |
| MLP | 35.60 | 4.06 | 0.26 | 0.02 | 0.26 | 0.02 | 0.26 | 0.02 |
| RF | 40.00 | 3.50 | 0.30 | 0.02 | 0.30 | 0.02 | 0.30 | 0.02 |
| DT | 40.00 | 4.83 | 0.38 | 0.07 | 0.47 | 0.11 | 0.41 | 0.08 |
| | | | Group 2: Students with disabilities | | | | | |
| SVM | 24.00 | 10.75 | 0.18 | 0.23 | 0.24 | 0.14 | 0.16 | 0.13 |
| MLP | 36.00 | 8.43 | 0.33 | 0.14 | 0.30 | 0.08 | 0.30 | 0.10 |
| RF | 40.00 | 14.14 | 0.43 | 0.20 | 0.41 | 0.19 | 0.38 | 0.16 |
| DT | 30.00 | 6.67 | 0.22 | 0.10 | 0.25 | 0.11 | 0.58 | 1.20 |
| | | | Group 3: Local students | | | | | |
| SVM | 51.00 | 0.00 | 0.46 | 0.00 | 0.26 | 0.00 | 0.24 | 0.00 |
| MLP | 48.80 | 1.23 | 0.38 | 0.03 | 0.31 | 0.01 | 0.32 | 0.01 |
| RF | 53.10 | 1.97 | 0.46 | 0.04 | 0.32 | 0.01 | 0.33 | 0.01 |
| DT | 45.20 | 1.23 | 0.35 | 0.01 | 0.35 | 0.02 | 0.35 | 0.01 |

*4.2. Results of Case 2: Minority Focus and Combination (MFC)*

Table 3 summarizes the classification results of Case 2. In this case, we only focus on two classes, EX and Poor. From this table, we find that the accuracy in three groups was improved. And, PR, Rec, and F1 are all acceptable. In Group 1 and 2, SVM can fully predict 100% EX and Poor examples. However, in Group 3 (local students), RF can have the best performance. Table 4 shows the confusion matrix results. It is obvious that RF can identify the Poor class very well (0.931), but it poorly classifies the EX class (0.412) for local students. Therefore, we implemented random oversampling in Case 3.

**Table 3.** Classification results of Case 2 (Minority focus and combination (MFC)).

| Performance | Accuracy (%) | | Precision | | Recall | | F1-Score | |
|---|---|---|---|---|---|---|---|---|
| | Mean | SD | Mean | SD | Mean | SD | Mean | SD |
| Methods | | | Group 1: International students | | | | | |
| SVM | 80.00 | 26.67 | 0.70 | 0.39 | 0.80 | 0.26 | 0.73 | 0.35 |
| MLP | 92.00 | 13.98 | 0.94 | 0.10 | 0.93 | 0.12 | 0.92 | 0.14 |
| RF | 90.00 | 17.00 | 0.94 | 0.11 | 0.91 | 0.15 | 0.89 | 0.88 |
| DT | 94.00 | 9.66 | 0.95 | 0.08 | 0.94 | 0.10 | 0.94 | 0.10 |
| | | | Group 2: Students with disabilities | | | | | |
| SVM | 100.00 | 0.00 | 1.00 | 0.00 | 1.00 | 0.00 | 1.00 | 0.00 |
| MLP | 96.30 | 11.10 | 0.94 | 0.17 | 0.93 | 0.22 | 0.93 | 0.20 |
| RF | 96.30 | 11.10 | 0.94 | 0.17 | 0.93 | 0.22 | 0.93 | 0.20 |
| DT | 96.30 | 11.10 | 0.94 | 0.17 | 0.93 | 0.22 | 0.93 | 0.20 |
| | | | Group 3: Local students | | | | | |
| SVM | 88.10 | 3.54 | 0.84 | 0.07 | 0.71 | 0.07 | 0.74 | 0.07 |
| MLP | 87.50 | 5.87 | 0.81 | 0.07 | 0.75 | 0.06 | 0.77 | 0.07 |
| RF | 92.10 | 2.02 | 0.89 | 0.04 | 0.81 | 0.07 | 0.84 | 0.06 |
| DT | 85.60 | 3.84 | 0.75 | 0.08 | 0.77 | 0.08 | 0.76 | 0.08 |

**Table 4.** Confusion matrix results of Case 2.

| | True Positive Rate | False Negative Rate | False Positive Rate | True Negative Rate |
|---|---|---|---|---|
| (a) Group 1: International students (SVM) | 1 | 0 | 0 | 1 |
| (b) Group 2: Students with disabilities (SVM) | 1 | 0 | 0 | 1 |
| (c) Group 3: Local students (RF) | 0.412 | 0.588 | 0.069 | 0.931 |

Note: Positive: "EX"; Negative: "POOR".

### 4.3. Results of Case 3: Random Oversampling the Minority (ROM)

Table 5 lists the classification results of Case 3. Compared to Case 1 and Case 2, it is clear that the class imbalance problem has been tackled. In Group 1 (international students), SVM, MLP, and DT could achieve 100% performance in both accuracy and F1. For Group 2 (students with disabilities), SVM can fully predict both Poor and EX examples. In Group 3 (local students), RF outperforms the other 3 methods in ACC, PR, Rec, and F1. Consequently, we will use the results of Case 3 to select the important features associated with students' learning performance.

**Table 5.** Classification results of Case 3 (Random oversampling the minority (ROM)).

| Performance | Accuracy (%) | | Precision | | Recall | | F1 | |
|---|---|---|---|---|---|---|---|---|
| | **Mean** | **SD** | **Mean** | **SD** | **Mean** | **SD** | **Mean** | **SD** |
| Methods | | | | Group 1: International students | | | | |
| SVM | 100.00 | 0.00 | 1.00 | 0.00 | 1.00 | 0.00 | 1.00 | 0.00 |
| MLP | 100.00 | 0.00 | 1.00 | 0.00 | 1.00 | 0.00 | 1.00 | 0.00 |
| RF | 96.60 | 7.17 | 0.97 | 0.06 | 0.97 | 0.06 | 0.97 | 0.07 |
| DT | 100.00 | 0.00 | 1.00 | 0.00 | 1.00 | 0.00 | 1.00 | 0.00 |
| | | | | Group 2: Students with disabilities | | | | |
| SVM | 100.00 | 0.00 | 1.00 | 0.00 | 1.00 | 0.00 | 1.00 | 0.00 |
| MLP | 96.00 | 12.65 | 0.98 | 0.08 | 0.97 | 0.10 | 0.96 | 0.13 |
| RF | 98.00 | 6.32 | 0.98 | 0.05 | 0.98 | 0.05 | 0.98 | 0.06 |
| DT | 94.00 | 13.50 | 0.96 | 0.09 | 0.95 | 0.11 | 0.94 | 0.14 |
| | | | | Group 3: Local students | | | | |
| SVM | 91.90 | 1.45 | 0.92 | 0.02 | 0.92 | 0.01 | 0.92 | 0.02 |
| MLP | 94.90 | 0.74 | 0.95 | 0.01 | 0.95 | 0.01 | 0.95 | 0.01 |
| RF | 98.60 | 0.52 | 0.99 | 0.01 | 0.99 | 0.01 | 0.99 | 0.01 |
| DT | 95.20 | 0.79 | 0.95 | 0.01 | 0.96 | 0.01 | 0.95 | 0.01 |

Moreover, as shown in Table 6, the AUC results for RF, SVM, MLP, and DT models for Group 1 are 1.00, 1.00, 1.00, and 1.00; for Group 2 they are 1.00, 1.00, 0.83, and 1.00; and for Group 3 they are 1.00, 1.00, 0.96, and 0.95. An outperformed AUC score obtained from our proposed models showcases that the classification performance is better and acceptable. In Table 7, the outcomes of confusion matrices show that for Group 1 and Group 2, the True Positive Rate (TPR) value are all 1.00. For Group 3, the TPR value is 1.00 and the True Negative Rate (TNR) value is 0.934. It can be indicated all four models (SVM, RF, MLP, and DT) have excellent performance in Case 3.

**Table 6.** AUC results of Case 3.

| | **RF** | **SVM** | **MLP** | **DT** |
|---|---|---|---|---|
| (a) Group 1: International students | 1.00 | 1.00 | 1.00 | 1.00 |
| (b) Group 2: Students with disabilities | 1.00 | 1.00 | 0.83 | 1.00 |
| (c) Group 3: Local students | 1.00 | 1.00 | 0.96 | 0.95 |

**Table 7.** Classification results of Case 3.

| | **True Positive Rate** | **False Negative Rate** | **False Positive Rate** | **True Negative Rate** |
|---|---|---|---|---|
| (a) Group 1: International students | 1.00 | 0 | 0 | 1.00 |
| (b) Group 2: Students with disabilities | 1.00 | 0 | 0 | 1.00 |
| (c) Group 3: Local students | 1.00 | 0 | 0.066 | 0.934 |

Note: Positive: "EX"; Negative: "POOR".

## 5. Discussion

Most of the attention of resampling methods for imbalanced classification is put on oversampling the minority class. However, some techniques have been developed for under-sampling the majority class. In order to provide an in-depth discussion about the used re-sampling (ROM) method in the analysis, we implement the random under-sampling technique. Table 8 lists the comparison of under-sampling, ROM and MFC. In this table, MFC represents the original data without implementing re-sampling techniques. From this table, we can reach three conclusions. First, oversampling (ROM) outperforms under-sampling, no matter the accuracy and F1. Second, a skewed class distribution might not necessarily cause class imbalance problems. As can be seen from Table 4, Group 1 and 2 do not have a class imbalance problem. Group 3 has a class imbalance problem, since the classifier gains a poor accuracy rate for minority class, but a higher accuracy rate for majority class. Third, if we implement re-sampling methods for those data without having class imbalance problems (Group 1 and 2), the classification performance cannot be improved. Therefore, it is recommended that future researchers not only use the re-sampling technique just by looking at the distribution class, but also look at the classification performance, whether there is a very high detection rate for majority class examples, or a very low classification rate for minority class examples.

**Table 8.** Comparison of re-sampling techniques (under-sampling and ROM).

| Performance | Under-sampling | | ROM | | MFC | |
|---|---|---|---|---|---|---|
| | Accuracy | F1 | Accuracy | F1 | Accuracy | F1 |
| Methods | | | Group 1: International students | | | |
| SVM | 86.63 | 0.86 | 100.00 | 1.00 | 80.00 | 0.73 |
| MLP | 83.30 | 0.82 | 100.00 | 1.00 | 92.00 | 0.92 |
| RF | 96.70 | 0.97 | 96.60 | 0.97 | 90.00 | 0.89 |
| DT | 93.40 | 0.93 | 100.00 | 1.00 | 94.00 | 0.94 |
| | | | Group 2: Students with disabilities | | | |
| SVM | 0.00 | 0.0 | 100.00 | 1.00 | 100.00 | 1.00 |
| MLP | 20.00 | 0.20 | 96.00 | 0.96 | 96.30 | 0.93 |
| RF | 40.00 | 0.40 | 98.00 | 0.98 | 96.30 | 0.93 |
| DT | 0.00 | 0.00 | 94.00 | 0.94 | 96.30 | 0.93 |
| | | | Group 3: Local students | | | |
| SVM | 86.60 | 0.86 | 91.90 | 0.92 | 88.10 | 0.74 |
| MLP | 85.30 | 0.85 | 94.90 | 0.95 | 87.50 | 0.77 |
| RF | 87.70 | 0.87 | 98.60 | 0.99 | 92.10 | 0.84 |
| DT | 77.40 | 0.76 | 95.20 | 0.95 | 85.60 | 0.76 |

In practice, the prediction models built in Case 3 are more meaningful than the models of Case 1 and Case 2. Therefore, we focused on the results of Case 3: ROM. We used the Gini index of the SVM algorithm to select important features associated with Group 1 learning performance and that of the RF algorithm used for the learning performance of Group 2 and Group 3. Figure 6 graphically displays the ranking of 15 input features and Table 9 lists the seven top-ranking important features associated with learning performance of three groups of students.

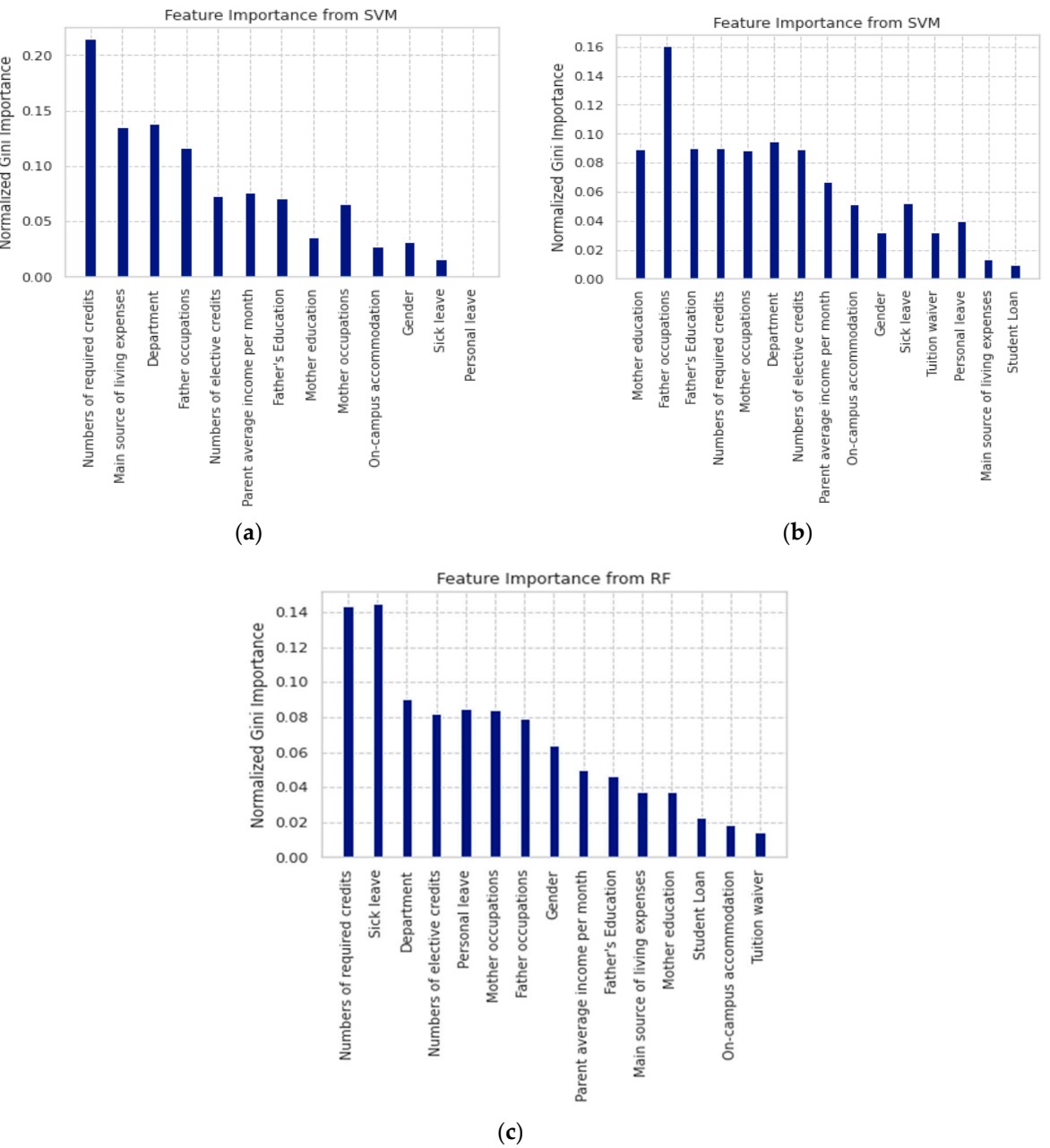

**Figure 6.** Gini feature importance: (**a**) Group 1: from SVM; (**b**) Group 2: from SVM; (**c**) Group 3: from RF.

**Table 9.** The seven top-ranking important features associated with learning performance of three groups of students.

| Group 1: International Students | Group 2: Students with Disabilities | Group 3: Local Students |
| --- | --- | --- |
| 1 No. of required credits | 1 Father occupations | 1 No. of required credits |
| 2 Department | 2 Department | 2 Sick leave |
| 3 Main source of living expenses | 3 Mother education | 3 Department |
| 4 Father occupations | 4 No. of required credits | 4 Personal leave |
| 5 Parent average income per month | 5 No. of elective credits | 5 Mother occupations |
| 6 Numbers of elective credits | 6 Father education | 6 Numbers of elective credits |
| 7 Father's education | 7 Mother occupations | 7 Father occupations |

For international students, the seven top-ranking features include [numbers of required credits], [main source of living expenses], [department], [father occupations], [numbers of elective credits], [parent average income per month], and [father's education]. It is clear that financial issues seem very important to international freshmen students since they live far away from their homeland. In addition, due to the Chinese language barrier and different cultures, it is very difficult for international students to find part-time jobs when they first arrive in Taiwan. Thus, campuses should offer them financial supports, e.g., scholarships and tuition waiver since Scholarships and tuition waiver are considered one of the contributing features for expanding international enrollment [5,55]. Apart from finances, numbers of required and elective credits contribute to both excellent and poor academic performance. As newcomers in the first semester, students find that learning and teaching methods, academic requirements, materials, and language quite different from those in their home countries. They do not know how many credits are suitable for their study. The unique relation with schools is through the international affairs department and tutors in academic departments. However, in practice, some departments receive international students for the first time, thus, they do not know how to assist international students' learning. In addition, intensive courses in the Mandarin language are offered to international students. Nevertheless, in the first semester, students' Chinese level is not adequate for long lectures and challenging assignments. Some departments offer teaching assistants and/or language supporters; whereas the other departments do not. Therefore, students studying in the departments with language supporters may have excellent performance, and vice versa, students studying in the departments without language assistance may have poor academic performance. Thus, schools should offer orientation/guided sessions and language supporters at the first semester. Academic staff should select understandable and comprehensive materials to ease students' learning process. Teaching methods should be adaptable to the students' needs. In contrast, the four factors [on-campus accommodation], [gender], [sick leave], and [personal leave] had the least effect on their learning performance.

For students with disabilities, the seven top features are [father occupations], [mother education], [father education], [numbers of required credits], [mother occupation], [department], and [numbers of elective credits]. As students need special assistance, families play an essential role in both their daily and their academic lives. Universities should involve families in student learning. For instance, they could allow family members to enter the classes just in case students need helps since the family realize the students' specific health status. By contrast, the four features: [tuition waiver], [personal leave], [main source of living expenses], and [student loan] had the lowest impact.

For local students, the seven top-ranking features consist of [numbers of required credits], [sick leave], [department], [personal leave], [mother occupations], [numbers of elective credits], and [father occupations]. Sick and personal leaves seem more important than the other features. Leaving school may not be the main reason for laziness, but students may have difficulties in their learning process as newcomers. In the first semester, students experience new academic lives, thus they need time for adjustment. In addition, family background is important to the local students' learning performance. Nevertheless, the four features: [mother education], [student loan], [on-campus accommodation], and [tuition waiver] hardly affected their academic performance.

## 6. Conclusions

The present study successfully built prediction models for the academic performance of international students and students with special needs at a Taiwanese vocational and technical university using the SVM, MLP, RF, and DT algorithms. The findings tackled the limitations of our previous study in [24]. The experimental results showed that for international students, three models: SVM (100%), MLP (100%), and DT (100%) were significantly superior to RF (96.6%); for students with disabilities, SVM (100%) outperformed RF (98.0%), MLP (96.0%), and DT (94.0%); for local students, RF (98.6%) outperformed

DT (95.2%) MLP (94.9%), and SVM (91.9%). The most important features were [numbers of required credits], [main source of living expenses], [department], [father occupations], [mother occupations], [numbers of elective credits], [parent average income per month], and [father education]. The outcome of this study is expected to benefit stakeholders in similar educational contexts when offering studying programs to international students and students with special needs.

Teachers can adapt more flexible teaching methods and offer understandable and comprehensive materials to facilitate students' learning. Orientation/guided sessions and tutoring should be offered to minority students. Availability and accessibility of scholarships should be offered to international students in order to expand international enrollment and maintaining student retention. The government can establish educational policies from the analysis of cross-institutional data. Parents can solve the students' problems, especially financial problems and guide them properly.

The major contributions of this study could be summarized as follows.

(1). To provide a predictive model for early warning for the academic performance of students with special needs and international students in universities.

Previous studies used variable data obtained during the semesters, such as quizzes, homework, absenteeism, etc. to predict academic performance. Our study only uses family factors, departments, and numbers of credits to predict minority students' performance before students start the first semester. This early prediction can support HEIs in providing additional assistances and proposing preventive measures prior to the start of the semesters

(2). To determine key factors associated with the academic performance of students with special needs and international students in Taiwanese technical and vocational universities.

In Taiwan, students with special needs have a high dropout rate and generally poor academic performance. However, in accordance with the special education laws, these students have educational funds provided by the Taiwanese government to assist their studies. Our study allowed Taiwanese HEIs to identify the students who really need help in order to enhance the counseling for students with special needs.

(3). To realize the differences between students with excellent academic performance (international students) and students with poor academic performance (students with special needs).

In Taiwanese technical and vocational universities, international students need outstanding academic performance to win scholarships based on a stricter screening mechanism; therefore, their academic performance is excellent. Students with special needs, on the other hand, are generally less effective in learning because of congenital physical and psychological limitations. This study assists technical and vocational HEIs to distinguish the important factors associated with the academic performance of outstanding students (international students) and poor students (students with special needs). This study also suggests some measures for educational teams to expand international enrollment and maintain student retention. For future research, other EDM approaches can be used to predict the minority students' performance and compare them to the outcomes of this study. Other influencing features, such as admission criteria (i.e., entrance exam scores and high school GPA) and educational background should be considered and recommended for future research in order to better predict applicants' future academic performance before admission.

**Author Contributions:** Conceptualization, T.-T.H.-C. and L.-S.C.; methodology, T.-T.H.-C.; software, T.-T.H.-C.; validation, T.-T.H.-C. and K.-V.H.; formal analysis, T.-T.H.-C.; writing—original draft preparation, T.-T.H.-C.; writing—review and editing, L.-S.C.; visualization, K.-V.H.; supervision, L.-S.C.; project administration, L.-S.C.; funding acquisition, L.-S.C. All authors have read and agreed to the published version of the manuscript.

**Funding:** This research was funded in part by the National Science and Technology Council, Taiwan (Grant No. MOST 111-2410-H-324-006).

**Institutional Review Board Statement:** Not applicable.

**Informed Consent Statement:** Not applicable.

**Data Availability Statement:** Not applicable.

**Conflicts of Interest:** The authors declare that they have no conflict of interest.

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
