# Peer review of "Learning Performance of International Students and Students with Disabilities: Early Prediction and Feature Selection through Educational Data Mining"

_2504-2289, doi:10.3390/bdcc6030094_

Round 1

Reviewer 1 Report

Data-driven analysis and machine learning algorithms have been widely adopted in the Education sector. In this paper, authors attempted to build prediction models for learning performance of international students and students with disabilities, using MLP, SVM, RM and DT based on the students’ profile and their GPA results. Overall, the topic of the study is important, but the analysis and discussion are too brief. Several major and minor issues are listed below.

1.     The Abstract is unclear. It needs to be rewritten in a more consistent way. For example, 1) it’s hard to see how the following are connected as COVID-19 started from 2019 but the data for the increase of students with disabilities are for year 2001-2015:  “In Taiwan, the incoming international students remain unchanged owing to the COVID-19 pandemic. However, the number of disable students increased in various levels between 2001-2015”; 2) what are the findings/outcomes/results from the study? 

2.     Page 1, Line 37, “educational higher institutions (HEIs)”, should it be ‘Higher Education Institutions (HEIs)’?

3.     Page 2, Line 65-66, what is the scope coverage of the “budget for special education”? Was it dedicated to support students with disabilities only? 

4.     In Section 3.2, information contained in Line 252-254 overlaps with Figure 2.

5.     There are several major issues with experimental results (Section 4). 

a.     In Figure 6-9, the authors didn’t mention which algorithm(s) was used to generate the results, i.e., precision, recall, f1-score, etc.

b.     In Section 4.1, the authors concluded that “the imbalanced class issue occurred in Case 1”, however, since the sample sizes (for international students and the students with disabilities) of Group 1 & 2 are extremely small, it’s unclear whether the results were due to the sample size or/and the “imbalanced class”. The same question is for the Case 2 analysis. 

c.     My most concern is with the use of ROM whether it is appropriate. I strongly recommend the authors to provide solid analysis and theoretical foundation to support the use of ROM in the context. 

6.     In Figure 11, the title of Figure 11(b) (“Feature Importance from RF”) is different from the description given in the Caption (“Group 2: from SVM”). More importantly, the authors need to explain the findings rather than listing some descriptive information. 

7.     It is hard to see how the conclusions are linked to the analysis. For example, in Section Discussion and conclusion (Page 15, Line 451-453), “… without language assistance, students have poor academic performance”, however this is not shown in the “top 7-ranking” important features identified from the machine learning algorithms (as shown in Table 2).  

Reviewer 2 Report

1 – Please replace ‘disabled students’ by ‘students with special needs’ or similar to be politically correct.

2 - The contributions of the paper are unclear. Please make the contributions in bullets

3 - There are several English mistakes. The manuscript needs revision by an English editor

4 - line 138, “The SLPP methods are classified into two main streams: supervised and unsupervised,” there is no information or any example about unsupervised learning. Please provide some

5 – Section 2.4, what is the rationale of selecting these classifiers (MLP, DT, SVM, RF) and not other classifiers

6 – line 246, what do you mean by “After deleting missing, irrelevant, noisy, and/or inconsistent data” how do you define this? The meaning of missing values is clear but what about others?

7 – line 257, “Out of 24 initial features (Section 3.1), a total number of 16 features were finally selected” how feature selection was conducted?

8 – lines 352 to 354 “The code for these metrics used in Python language are available at

https://scikit-learn.org/stable/modules/classes.html#module-sklearn.metrics and

https://www.youtube.com/watch?v=prWyZhcktn4&t=904s.” this information is not significant

9 – Why the GPA was treated as nominal value and not kept as numeric so that the problem will be treated as regression problem instead of classification?

10 – screenshots such as figures 6, 7 and 9 should be removed and replaced by proper tables

11 – There is no comparison with related work

Reviewer 3 Report

Figure 6, Figure 7 and Figure 9 can be provided as a Table.

Hyperparameter optimization done?

conclusion section can be made as a separate section.

Results should be summarized in table rather than showing in screenshot.

Comparision with existing studies required.

Highlight the contributions of the proposed work.

Round 2

Reviewer 1 Report

The authors have addressed most of my concerns raised the previous version of the manuscript. Although the authors provided some references regarding the use of the ROM method in the experiment and analysis, but it still remain as a major concern whether the use of the method is appropriate in this context (with a sample of 1, 2, 3, or 4). I strongly recommend the authors to provide a in-depth discussion about the use of the method in the analysis. 

Reviewer 2 Report

The authors addressed my comments and I am happy to accept the paper

Author Response

Responses to Reviewer #2

Thank you for your comments regarding the submitted paper. 

Comment #1

The authors addressed my comments and I am happy to accept the paper.

Response:

We appreciate your great comments and acceptance.

Round 3

Reviewer 1 Report

The authors have provided sufficient explanations to address the concerned raised in the previous version. I believe it is now in a good form for publishing the manuscript in the journal.